# Pomegranate Woody Mycobiota Associated with Wood Decay

**DOI:** 10.3390/jof11040254

**Published:** 2025-03-26

**Authors:** Valentino Bergamaschi, Maria Teresa Valente, Rosario Muleo

**Affiliations:** 1Research Centre for Plant Protection and Certification (CREA-DC), Council for Agricultural Research and Economics (CREA), 00156 Rome, Italy; mariateresa.valente@crea.gov.it; 2Department of Agriculture and Forest Sciences (DAFNE), University of Tuscia, 01100 Viterbo, Italy; muleo@unitus.it

**Keywords:** wood decay, *Diaporthe*, pathogenicity test, metabarcoding, ITS 2

## Abstract

The rapid expansion of pomegranate (*Punica granatum* L.) cultivation in central and southern Italy has revealed emerging phytosanitary challenges, including “pomegranate wood decay syndrome”, characterised by cortical cankers, wood browning, and progressive plant decline. This study investigates the fungal microbiota associated with symptomatic pomegranate wood using a combined approach of traditional fungal isolation and ITS2 metabarcoding analysis. Samples from two orchards in Lazio were examined, revealing a complex fungal community with a high prevalence of *Neofusicoccum parvum* (putative) and species belonging to the genus *Diaporthe*. Pathogenicity tests confirmed the role of *N*. *parvum* in causing significant wood browning, while other isolates showed variable virulence. Statistical analyses validated the pathogenicity of select isolates, with the putative *Diaporthe eres* (Nitschke) consistently demonstrating potential pathogenic activity across all trials. Metabarcoding identified 289 taxa, highlighting a richer fungal diversity in the symptomatic wood compared to the asymptomatic sections. Notably, *Coniella granati*, previously implicated in pomegranate decline, was absent in the studied orchards. The findings reveal that pomegranate wood decay is a complex syndrome driven by fungal pathogens and environmental stressors, such as low temperatures. This study highlights the value of integrative approaches for understanding and managing fungal-associated wood diseases in pomegranate orchards.

## 1. Introduction

The interest in the cultivation of pomegranates (*Punica granatum* L.) in Italy has led to research to increase the knowledge about this crop, whose fruit is considered a functional product of great benefit from the organoleptic and nutraceutical points of view, as well as for the possibility of pharmaceutical applications, given the content of active molecules that reduce the risk of vascular diseases, coronary artery problems, and cancer mortality [1,2]. Among the bioactive compounds present in pomegranate juice, there are phenols and tannins such as punicalin, punicalagin, and ellagic acid; moreover, varieties with red arils contain a large number of anthocyanins, including cyanidin, delphinidin, and pelargonidin, and related glycosides with strong antioxidant capacities [3,4,5].

In some of the main pomegranate cultivation areas of Italy, including the central and southern regions as well as the islands, young plants have experienced phytosanitary problems that, in some cases, have led to the loss of both fruit production and entire plants [6]. This disease, called “Pomegranate canker” or, more generically, “Pomegranate decay”, shows progressively worsening conditions of the plant: the first symptom is leaf chlorosis, and later, the plant shows the formation of cankers on the stem and branches with wood browning below the symptoms [6]. The main fungi isolated from the symptomatic portions of pomegranate wood were *Coniella granati*, previously reported in Italy, Iran, Israel, Turkey, Greece, and Spain [6,7,8,9,10,11], and *Neofusicoccum parvum* (Pennycook and Samuels) Crous, Slippers, and Phillips, previously reported in Greece [12], in Florida [13], and recently in an orchard in Lazio, Italy [14]. *Neofusicoccum parvum* was also reported on other plants of agricultural interest, including the grapevine, kiwifruit, and avocado [15,16,17]. This fungus enters through wounds (e.g., pruning cuts or mechanical damage due to abiotic agents such as frost or hail). The infection causes the browning of the bark on the trunk and branches and the discolouration of the wood underlying the affected areas, resulting in the general decay of the plant and subsequent die-off of the shoots. The fungus often causes significant cankers that can affect the whole plant [14].

Wood decay often represents a disease complex, due to the characteristic mixed infections [18], and the simultaneous presence of several fungal species complicates the diagnosis. The traditional phytosanitary approach requires an initial culturing step, which excludes the detection of biotrophic, endophytic species and may limit the detection of slow-growing fungi. As trunk pathogens cause mixed infections, it is important to characterise the composition of the species complex. In recent years, due to the growing power and reduced cost of next-generation sequencing technologies, DNA metabarcoding analysis has been used extensively for the characterisation of microbial communities [19,20,21,22,23]. The internal transcribed spacer (ITS) regions of ribosomal DNA (rDNA) are the most predominant DNA barcode sequences used for fungal metabarcoding, since they can be easily amplified and sequenced and they are highly represented in GenBank and other databases [24,25]. The Illumina MiSeq NGS technology is the most used in metabarcoding studies [26], due to its high diversity coverage, high accuracy, low cost, and availability of several consolidated data processing pipelines; however, due to the limited sequence lengths, it allows for the sequencing of only one of the ITS subregions. The choice of using either ITS1 or ITS2 for the best taxonomic resolution is still debated [27,28]. ITS2 shows some advantageous features: it is less variable in length, has no intron regions, is better represented in public databases, and has a longer portion of the sequence providing taxonomic information [29].

This work aims to study and characterise the fungal flora present in diseased pomegranate wood through a traditional phytosanitary analysis combined with an ITS metabarcoding analysis, and to verify the disease involvement of the more frequently isolated fungi through a pathogenicity test. To the best of our knowledge, this is the first study combining culturing and DNA metabarcoding approaches to assess the fungal pathogen diversity associated with pomegranate wood decay.

## 2. Materials and Methods

### 2.1. Collection of Samples

Two pomegranate Italian orchards, both in the Lazio region, were selected: Sermoneta (41.556459, 12.979830) and Pavona (41.742874, 12.603806). In both orchards, the pomegranate plants belong to the cultivar ‘Wonderful’. In Pavona, the soils are of volcanic origin, very deep, sub-acidic, non-calcareous, and characterised by a medium to fine texture, with good organic matter content. On the other hand, in Sermoneta, the substrate consists of alluvial sediments with a fine texture, featuring a surface mineral horizon highly enriched in organic matter and affected by limited oxygen availability. Overall, both soils show good organic matter contents [30], and according to the soil analysis provided by the owner, no deficiencies in macro- or micronutrients were detected. The soils of the Sermoneta orchard were classified predominantly as Haplic Luvisols (>75%) with Luvic Phaeozems (10–25%), while the Pavona orchard soils were classified as Haplic Phaeozems (25–50%), Luvic Phaeozems (10–25%), and Cambic Endoleptic Phaeozems (<10%) [30]. Furthermore, the Lazio region experiences colder winter climatic conditions than in southern Italy, where pomegranate cultivation is more successful [31]. Fifteen symptomatic plants obtained from both of the orchards from 2017 to 2019 were transferred to the laboratory of the Research Centre for Plant Protection and Certification (CREA-DC, Roma) and analysed by the isolation and morphological and molecular identification of the fungal flora. Two plants showing the symptoms of pomegranate decay (identified as Plant 1 and Plant 2, Figure 1) from the Pavona site (2018) were analysed by the conventional method (isolation and identification) and by the metabarcoding approach. Plant 1 presented some cankers on the stem and some branches, subcortical browning of the wood, and yellowing; Plant 2 showed an extended canker on the whole trunk, with extended subcortical browning (Figure 1).

The plants (stem and branches) were divided into sections (Figure 2) representative of the different asymptomatic and symptomatic wood tissues from the trunks and the branches.

### 2.2. Fungal Isolation and Identification

Longitudinal sections of the tissues sampled as described above were obtained aseptically; small portions between the healthy and symptomatic tissues were explanted from the browned wood and placed on a Potato Dextrose Agar (PDA) culture medium, amended with 100 µg/mL of streptomycin and ampicillin sodium salt antibiotics, and incubated at 22 °C until the development of fungal colonies. The colonies were then transferred onto fresh PDA plates and incubated at 22 °C under near-ultraviolet (NUV) light with a 12 h light/12 h dark photoperiod to promote fungal growth and sporulation. The fungal identification was carried out based on the morphological features of the colonies, including macroscopic characteristics such as colony colour, texture, and growth pattern, as well as microscopic observations of the fruiting structures, including pycnidia and conidiomata produced in the culture. The morphological identification was confirmed by the sequencing of the ITS region amplified with ITS4/ITS5 primers [32] performed at the BioFab Research laboratory (Rome, Italy). Since only the ITS region was used for molecular identification, isolates were classified as putative species based on the most abundant species showing 100% identity in the NCBI database. The sequences were blasted in the GenBank database by using the nBLAST search. The isolated and identified species were preserved in the fungal collection of the CREA-DC (Rome, Italy).

### 2.3. DNA Extraction

The DNA from the fungal mycelium was extracted with the rapid protocol of Cenis [33], starting from 100 mg of mycelium modified for the plate-grown mycelium [34]. The pomegranate wood was ground in liquid nitrogen; an aliquot of ground wood (about 100 mg) was used for DNA extraction, performed with the DNeasy Plant Mini Kit (QIAGEN, Hilden, Germany) according to the manufacturer’s instructions. The quantity and quality of the purified DNA extracts were determined using a Qubit 2.0 fluorometer (Life Technologies, Carlsbad, CA, USA) and a DS-11 FX spectrophotometer (DeNovix Inc., Wilmington, DE, USA), respectively.

### 2.4. Metabarcoding Analysis

A metabarcoding analysis was carried out on Plant 1 and Plant 2 from the Pavona pomegranate orchard to confirm the data derived from the isolation and identification of the fungal flora, and to obtain more information about the fungal communities in the asymptomatic and symptomatic wood tissues. The different samples were combined into 8 experimental units (Table 1), 7 from Plant 1 and 1 from Plant 2.

The DNA samples were amplified by PCR targeting the ITS2 region of the fungal rDNA using ITS3 and ITS4 primers [32]. These primers were modified by the 5′-end Illumina overhang adapter sequences (34 and 35 nucleotides) and by the different 5′-end identifier (index) sequences of 6 nucleotides, generating 8 oligo pairs of unique tags (Appendix A). PCRs were conducted for the 8 samples with each of the 8 newly designed primer pairs to test for variability in the amplification success. Each sample was assigned a different forward/reverse index combination for sample-specific labelling.

The amplification reaction was performed following the protocol proposed by Miller et al. [35]. The reaction mix contained 10 ng of template DNA, 1× BioTaq PCR NH_4_-based reaction buffer, 4 mM of MgCl_2_, 0.8 mM of dNTP mix (Bioline, London, UK), 0.8 mg/mL of bovine serum albumin (BSA, Invitrogen, Carlsbad, CA, USA), 2.5 U of BioTaqTM DNA polymerase (Bioline, London, UK), and 0.5 μM of each forward and reverse primer in a 25 μL reaction volume. For each sample, three PCRs were carried out at three different annealing temperatures; combining PCR products using a range of annealing temperatures reduces the primer bond distortion and decreases the stochasticity of the individual PCRs [19]. The following cycling conditions were used for the amplification: an initial denaturation at 94 °C for 30 s, followed by 30 cycles of denaturation at 94 °C for 1 min, annealing at 50/53/56 °C for 1 min, extension at 72 °C for 30 s, and a final extension at 72 °C for 2 min. The PCR products were visualised in 1% agarose gel electrophoresis using GelRed™ dye (Biotium, Hayward, CA, USA). Purification of the amplicons was performed using the Isolate II PCR and Gel Kit (Bioline, London, UK). The cleaned PCR products were then quantified using a Qubit 2.0 Fluorometer (Life Technologies, Carlsbad, CA, USA) and mixed in equimolar amounts (75 ng for each sample). A single sample consisting of the pooled amplification products was sent to the BioFab Research laboratory (Rome, Italy). The sequencing was performed on the Illumina MiSeq platform (Illumina, San Diego, CA, USA) with a 2 × 300 paired-end run and a reading depth of 100,000 reads. The reads forward and reverse were merged and filtered by quality (Q > 30). Subsequently, a demultiplexing of the sequenced sample was carried out, obtaining the results of the 8 separate experimental units. Bioinformatic processing of the sequences was conducted using the USEARCH pipeline and the UPARSE-OTU algorithm [36].

The paired-end (PE) sequences were merged using the -fastq_mergepairs command. Next, the PE reads were quality-filtered (maximum e-value: 1.0), trimmed to 340 bp, dereplicated, and sorted by abundance (singletons removed). Chimaera detection and operational taxonomic unit (OTU) clustering at 97% sequence identity followed. The original sequences were then mapped to OTUs at the 97% identity threshold, generating separate OTU tables for the prokaryotic and fungal communities.

The taxonomic affiliation of each OTU was assigned using the -syntax algorithm against the UNITE_all_eukaria database [37] with an 80% confidence threshold. The sequencing depth was normalised across the libraries, and the resulting OTU tables were used for downstream analyses.

### 2.5. Pathogenicity Test

Artificial inoculations were performed using the 5 fungal species isolated with a higher incidence and belonging to the most represented *Genus* in the metabarcoding analysis (*Neofusicoccum parvum*, isolate ER 2123; *Diaporthe eres* (Nitschke), isolate ER 2111; *D*. *phaseolorum* (Cooke & Ellis) Sacc., isolate ER 2126; *D. rudis* (Sacc.) Sacc., isolate ER 2127; *D*. *foeniculina* (Sacc.) Udayanga & Castl., isolates ER 2125 and ER 2128). *Neofusicoccum parvum* ER 2123 was used as a positive control, since it is reported as a pathogen of pomegranate wood [14].

The pathogenicity was evaluated through a test carried out on the cut twigs and plants. The twigs were prepared in sections approximately 20 cm long and harbouring at least two buds, between which were performed the inoculation. The inoculation area was previously surface sterilised with denatured ethyl alcohol.

Three inoculation procedures were performed.

First procedure: Detached pomegranate twigs at least one year old, as described by Palavouzis et al. [12]. The test was carried out with 5 mm diameter mycelium plugs taken from the margins of the fungal colonies, about one week old, grown in purity on the PDA medium. A 5 mm diameter disc of the bark was previously removed from the twigs with a cork borer to expose the cambium to the mycelium applied with a PDA plug. Then, the twigs were placed in the dark in a humid chamber at room temperature to promote the infection (Figure 3). After 3 days, when presumably the mycelium had colonised the wood, the humid chambers were opened. The test was completed at 7 days post inoculation (DPI).

Second procedure: Detached twigs of at least four years old, placed in water, were inoculated, wounding the bark with a razor blade in a basipetal orientation and exposing the youngest wood; then, the strip of tissue was pulled back to place the PDA with the mycelium on the exposed xylem. The inoculation area was covered with a cotton disc soaked with 3 mL of sterile tap water, wrapped with tinfoil strips, and stuck with paper tape, as shown in Figure 4 [38]. The twigs were placed in water at 25 °C with a photoperiod of 13 h of light (13/11), and were checked weekly to change the water. After seven days, the tinfoil strips were removed. The test was completed at 90 DPI.

Third procedure. In vivo plants were inoculated as described in the second procedure and placed outdoors in the nursery. After 30 days, when presumably the mycelium had colonised the wood, the tinfoil strips and the paper tape were removed; the test was completed in 90 days.

At the end of these tests, the first layers of the bark were removed with a razor blade to show and measure the length of the browned area in the longitudinal direction of the twig (Figure 5).

The tests were conducted in randomised blocks, with four repetitions for procedures 1 and 2 and five repetitions for procedure 3. This was applied to each fungal isolate, as well as to the negative control (sterile PDA plugs) and the positive control (ER 2123).

Moreover, control isolations were carried out from the symptomatic areas to verify the Kock’s postulates.

### 2.6. Statistical Analysis

The data from the pathogenicity test were analysed using R software version 4.2.1 to perform a One-Way ANOVA, assessing the significant differences in browning length (cm) among the treatments to evaluate the virulence of the fungus compared to the negative controls.

Before the ANOVA, the normality was checked using the shapiro.test function, while the homoscedasticity and non-additivity were verified with the bartlett.test and leveneTest functions, respectively, considering *p* > 0.05 as the threshold for homogeneity. The ANOVA was performed using the aov function, followed by Tukey’s HSD post hoc test with the Tukey HSD function, to identify the significant differences between the treatment groups (*p* = 0.05). The ANOVA results were considered significant at *p* < 0.05 and highly significant at *p* < 0.01.

## 3. Results

### 3.1. Phytosanitary Analysis of Pomegranate Plants Affected by Decay

Plants exhibiting wood decay symptoms, including reduced growth, leaf yellowing, and reddening, were sampled. Their trunks and twigs displayed cortical cankers, with some located above the collar and others extending along the entire trunk. In the early stages of the disease, before the canker formation, the bark appeared darker, and the underlying cambium showed distinct darkening and damage.

Laboratory dissections revealed browned wood, particularly beneath the cankers, with necrosis spreading in both the basipetal and acropetal directions. In cases of multiple cankers, the browning could extend throughout the entire trunk.

The fungal isolations from 325 symptomatic tissues yielded 188 colonies. The morphological and DNA sequence analyses identified 11 distinct species (Table 2). *Neofusicoccum parvum* was detected at both the Pavona and Sermoneta sites, but only in early autumn. *Diaporthe* spp. and *Gliocladium* spp. were more frequently isolated and exhibited a high incidence.

### 3.2. Metabarcoding Analysis

A fragment of the expected size (340 bp) was obtained from the ITS2 amplification in all of the samples investigated.

The dataset comprises 114,486 reads distributed across the eight samples, with a total of 41 operational taxonomic units (OTUs) identified. Notably, all OTUs and samples contain nonzero counts, with no missing data or singleton OTUs.

In terms of OTU abundance and distribution, none of the OTUs have a total count of one, meaning that all identified taxa are represented by multiple reads across the samples. A total of 328 counts were recorded, with 44.5% (146 counts) being zero and 55.5% (182 counts) showing at least ten occurrences. This suggests a relatively balanced distribution of the microbial taxa, although a portion of the OTUs remain rare across the samples.

Regarding the core microbiomes and ubiquity of the OTUs, 13 OTUs (31.7%) are present in all samples, defining a core microbiome that is consistently detected across the dataset; 23 OTUs (56.1%) are detected in at least 50% of the samples, indicating a moderate level of species turnover across the different sampling points.

The sample sizes and sequencing depths vary, with the smallest sample containing 3749 reads and the largest 19,253 reads. The median sample size is 15,599 reads, while the mean is 14,310 reads, suggesting an overall well-balanced sequencing depth across the samples. The lower quartile (14,171 reads) and upper quartile (19,157 reads) indicate that most samples fall within a reasonable range, minimising the risk of uneven sequencing depth affecting the downstream analyses.

The NGS analysis of the ITS2 region revealed a wide diversity (Figure 6, Figure 7 and Figure 8).

Based on Figure 6, the most frequently occurring taxon across the samples was *Diaporthe*, which shows a consistently high relative abundance, ranging from 17.10% to 86.60%. This genus dominates several samples, particularly samples 1, 2, 5, and 6, where its relative abundance exceeds 60%.

Another notable taxon is *Punica*, which exhibits the highest relative abundance in sample 4 (91.30%) and sample 3 (54.50%), which were asymptomatic sections. In contrast, taxa such as *Vishniacozyma*, *Phaeoacremonium*, and *Fusarium* are present in lower proportions, but show some variability across the samples, with *Phaeoacremonium* reaching 10.70% in sample 7.

Regarding the species richness, sample 7 (symptomatic section) appears to have the highest diversity, as it contains multiple taxa with substantial relative abundances, including *Diaporthe* (41.80%), *Phaeoacremonium* (10.70%), *Fusarium* (5.44%), and *Clonostachys* (4.03%). This indicates a more heterogeneous microbial community in this sample.

Conversely, samples 3 and 4 (asymptomatic sections) exhibited the lowest species richness, as they are overwhelmingly dominated by *Punica* (54.50% and 91.30%, respectively) and *Diaporthe* (17.10% and 4.29%, respectively), with the other taxa either absent or present in very low amounts.

Overall, the results highlight a strong predominance of *Diaporthe*, variations in the microbial community structure across the samples, and a marked difference in the species richness, with sample 7 exhibiting the highest diversity and sample 4 the lowest.

The results confirmed the correlation between isolated fungal and fungal genera detected with metabarcoding analysis.

### 3.3. Pathogenicity Test

The artificial inoculation tests on the twigs were able to reproduce the symptom of browning of the wood by all tested fungi, which had been obtained from the diseased wood of pomegranate plants affected by decay, and to highlight their greater or lesser aggressiveness. The positive control *N. parvum* (ER 2123 tube) caused the expected symptoms in the three tests, confirming that the inoculation method worked properly (Appendix A). The ANOVA results highlighted highly significant differences in terms of the length of browning between the treatments in all three tests (Table 3, Table 4 and Table 5).

*Diaporthe eres* (ER 2111 tube) was the only isolate that showed a constant pathogenicity in the three analyses (Figure 9). The other isolates, *D. phaseolorum* (ER 2126 tube), *D. foeniculina* (ER 2128 and ER 2125 tubes), and *D. rudis* (ER 2127 tube), were less aggressive and showed no statistically significant differences compared to the control. Despite this, their presence on the symptomatic wood was not completely negligible, and although with less aggression, the mycelium managed to develop through the tissues of the pomegranate wood.

Figure 10, Figure 11 and Figure 12 show the average lengths (cm) of browning caused by the five inoculated isolates.

In the detached twig assay conducted under humid chamber conditions, significant differences in browning length were observed among the tested *Diaporthe* species (Figure 10). The control (PDA) treatment resulted in minimal browning, confirming the absence of natural infection. Among the isolates, *Diaporthe phaseolorum* (ER 2126) induced the most severe symptoms, causing significantly greater browning compared to the control and all other species. *Diaporthe eres* (ER 2111), *D. foeniculina* (ER 2125 and ER 2128), and *D. rudis* (ER 2127) caused intermediate levels of browning, with overlapping statistical groupings.

In the 2019 twig inoculation assay with an incision and incubation in water, significant differences in browning length were observed among the *Diaporthe* species tested, confirmed by Tukey’s HSD test (Figure 11). The control (PDA) exhibited minimal browning, confirming the effectiveness of the wounding method without fungal inoculation. *Diaporthe eres* (ER 2111) caused the most severe symptoms, producing significantly longer browning compared to all other treatments. *Diaporthe foeniculina* (ER 2125) also induced substantial browning, although less than *D. eres*, while *D. rudis* (ER 2127), *D. phaseolorum* ER 2126, and *D. foeniculina* ER 2128 showed intermediate levels of browning, with no significant differences compared to the control.

In the 2020 inoculation test on pomegranate plants (Figure 13a), significant differences in browning length were observed among the *Diaporthe* isolates (Figure 12 and Figure 13b). The control (PDA) treatment caused minimal browning (Figure 13c), confirming the absence of natural infection. Among the tested isolates, *Diaporthe eres* (ER 2111) induced the most severe symptoms, with significantly greater browning compared to all other treatments, according to Tukey’s HSD test. *Diaporthe rudis* (ER 2127) and *D. foeniculina* (ER 2128) caused intermediate browning lengths, statistically grouped together. *Diaporthe foeniculina* (ER 2125) and *D. phaseolorum* (ER 2126) showed lower levels of browning, not significantly different compared to the control. These results further confirm the high pathogenic potential of *D. eres* (ER 2111) and highlight the variability in aggressiveness among the *Diaporthe* species under field-like conditions.

## 4. Discussion

The rapid expansion of pomegranate cultivation in central and southern Italy, primarily using foreign varieties, has revealed unexpected phytosanitary issues. This is notable for a plant traditionally considered hardy and relatively free from phytosanitary concerns, apart from well-documented issues affecting the fruit. Over the past three years, the first cases of plant decline and mortality have been reported, with the losses reaching 30–40% in 2015 and escalating to 90% in subsequent years. In Puglia, in 2016, Pollastro et al. [6] identified *Coniella granati*, a pathogen already known in various pomegranate-growing regions worldwide, as the causative agent of fruit rot. However, in China, Iran, and Spain, *C*. *granati* has also been associated with cankers and pomegranate tree mortality, while in Greece and Turkey, it has been reported as a cause of stem cankers [39].

In central Italy, particularly in Lazio, this issue was initially unknown at the start of our investigation. However, following the reports and subsequent inspections and analyses described in this study, two potential fungal pathogens were identified: *N. parvum* (putative) and *Diaporthe* spp., which are likely to act synergistically. Initially, *N. parvum* was considered the only causal agent [14], and was therefore the primary focus of further investigations. The pathogenicity tests confirmed that the *Diaporthe* species are less aggressive than *N. parvum*, and likely play a minor role in symptom development. However, *Diaporthe* spp. were consistently isolated from the symptomatic wood and detected through the metabarcoding analysis, suggesting their constant presence in the affected tissues. The combined presence of *N. parvum* and *Diaporthe* spp. may contribute to symptom expression comparable to that caused by *C*. *granati*.

The metabarcoding analysis had limitations due to the availability of ITS2 sequences in the UNITE database, which, at present, does not allow for a taxonomic resolution sufficient to classify all DNA in the sample at the species level. As a result, a proportion of the fungal DNA could only be identified at the division or, at best, genus levels. Additionally, a significant proportion of the DNA (on average 29.3%) remained unassigned, potentially concealing other pathogenic fungi. The metabarcoding analysis, even within its limits, allowed us to highlight in the two plants analysed that *N. parvum* and *C. granati* were not present in the diseased wood, where it was not isolated, nor even in the healthy wood, excluding the possibility that it was present as an endophyte.

This result was significant, as it validated the test performed, confirming the presence of a substantially richer fungal community in the symptomatic tissue, which exhibited clear signs such as cankers and wood browning. In contrast, the fungal DNA detected in the asymptomatic samples was likely associated with the endophytic fungal flora naturally present in pomegranate wood.

Different diseases are already known as the Esca of Grapevine [40] and Esca-like symptoms of *Actinidia* [41] caused by fungi complexes. Today, there are other diseases, such as “Kiwifruit Vine Decline Syndrome” (KVDS), caused by different microorganisms and abiotic disorders. The causes of decline are still discussed; currently, pathogenic species such as *Phytopythium vexans* [42,43], *Phytophthora* spp. [44], *Cylindrocarpon* [45], and the anaerobic bacteria *Clostridium* are involved and reported [46].

Similarly to other complex syndromes, abiotic stress could represent a trigger factor for the disease. The pomegranate, mainly the ‘Wonderful’ cultivar, is widely cultivated in warm climates and originates from regions such as Iran and the surrounding regions, from where it spread to other parts of the world [47], where the temperature conditions are more similar to those of southern Italy rather than central regions like Lazio. This difference is attributed to the latitudinal gradient, as well as variations in altitude and proximity to the sea, which influence the mean winter temperatures [31] For this reason, climatic conditions play a central role as a trigger for wood decay.

Few studies have dealt with the taxonomic resolution obtained using both the ITS1 and the ITS2 barcodes on the same dataset [27,48,49,50,51]. They have been carried out on both *Ascomycota* [52,53] and *Basidiomycota* [54]. The choice of primers can introduce taxonomic bias, as they may produce a higher number of mismatches in certain taxa [53,55,56]. Some studies also reported that the two spacers are prone to preferential amplification at different levels [48,49,50,52,53]. Basidiomycetous taxa have, on average, longer amplicon sequences for ITS2, and since the shorter sequences are preferentially sequenced with high-throughput sequencing (HTS), the use of ITS2 would favour the detection of ascomycetes [53]. On the other hand, ITS1 often contains an intron that extends its sequence at the 5′-end [57], thereby promoting an over-representation of those sequences that lack the intron [49]. Because ITS2 is more frequently represented in public databases, has a higher number of available sequences, and offers a better taxonomic resolution, it has been proposed as the better choice for parallel sequencing [52]. In some cases, however, no substantial differences between ITS1 and ITS2 were recovered [27,54]. Finally, there are numerous studies that consider a single spacer, either ITS1 or ITS2 [35,53,58,59].

As fungal metabarcoding studies have used different HTS platforms [60], different bioinformatic pipelines have been proposed [61,62]. These have been developed based on the experience gained from the data analyses of prokaryote datasets [63]. However, no standard procedure has been established so far for fungal sequence data. The analyses seem strongly dependent on the working hypotheses of each study and on the type of sequence at hand. As the majority of the studies target fungal communities to uncover unknown diversity, an important and ongoing problem is the definition of those sequences lacking an assigned taxonomy [52]. For this reason, many sequences remain identified as “unassigned”. In addition, many fungi have not yet been sequenced and cannot offer reference sequences for ongoing studies [63].

The molecular markers used in this investigation have a broad level of fungal species identification, so the failure to detect *Coniella granati* among the identified species allows us to speculate that other species are implicated in pomegranate wood decay syndrome. In fact, the metabarcoding results suggested a relatively stable microbial community, with a substantial proportion of OTUs present in multiple samples. For *C. granati*, in the two plant objects of our study, it can be said with certainty that it was not present and, therefore, was not involved with the cankers and deaths found in Pavona and Sermoneta. This is confirmed both by the isolations carried out on the diseased plants and by the metagenomic analysis, where the presence of *Schizoparmaceae* was not found.

The presence of a well-defined core microbiome (31.7% of OTUs in all samples) indicates a consistent microbial signature across different conditions. The data also suggest a good sequencing depth, with minimal bias from zero- or low-count OTUs.

## 5. Conclusions

To conclude, pomegranate wood decay syndrome is probably a complex syndrome, where the triggering factor could be climatic conditions such as low temperature stress, from which plants become susceptible to various wood pathogens including *N. parvum* and other fungi of the *Diaporthe* complex. The wood deterioration caused by *C*. *granati* may represent a disease with its own epidemiology and progression, capable of damaging pomegranate plants even in the absence of abiotic stress.

## Figures and Tables

**Figure 1 jof-11-00254-f001:**
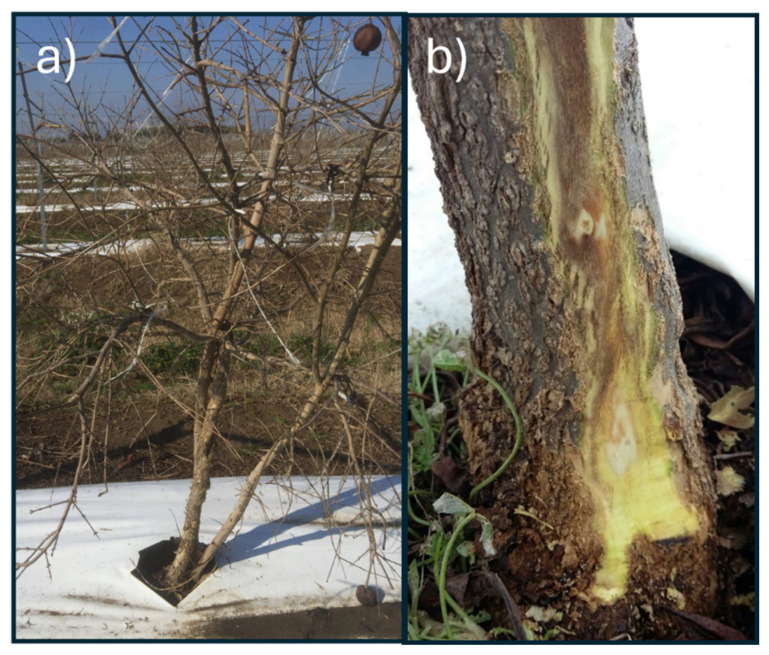
Symptomatic Plant 1 showing a large canker on the trunk (**a**) and subcortical browning on Plant 2 (**b**), both sampled for the metabarcoding analysis in Pavona (RM).

**Figure 2 jof-11-00254-f002:**
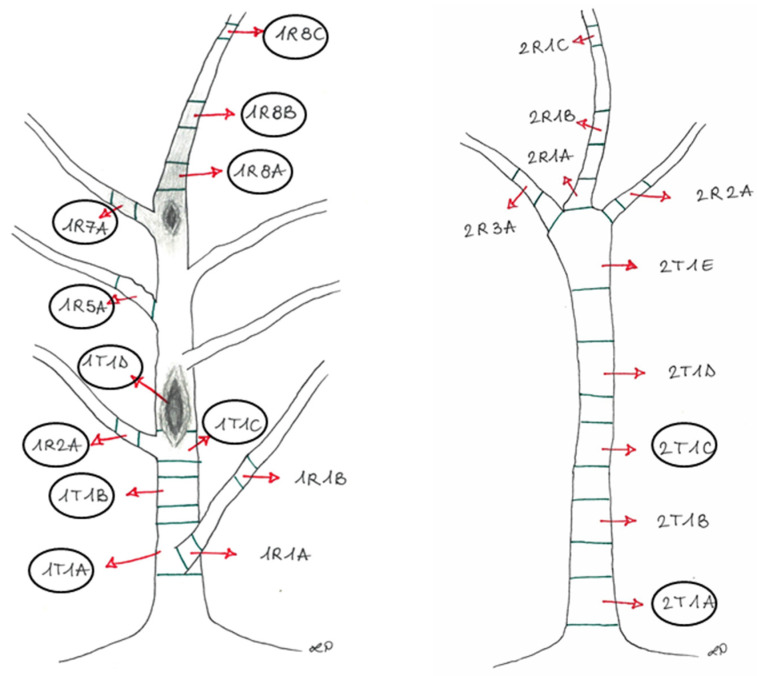
The sampling points of Plant1 (**left**) and Plant2 (**right**), showing the cankers and browning wood, with the identification legend of each sampled section, later described in Table 1. The circled sections were used for the metabarcoding analysis.

**Figure 3 jof-11-00254-f003:**
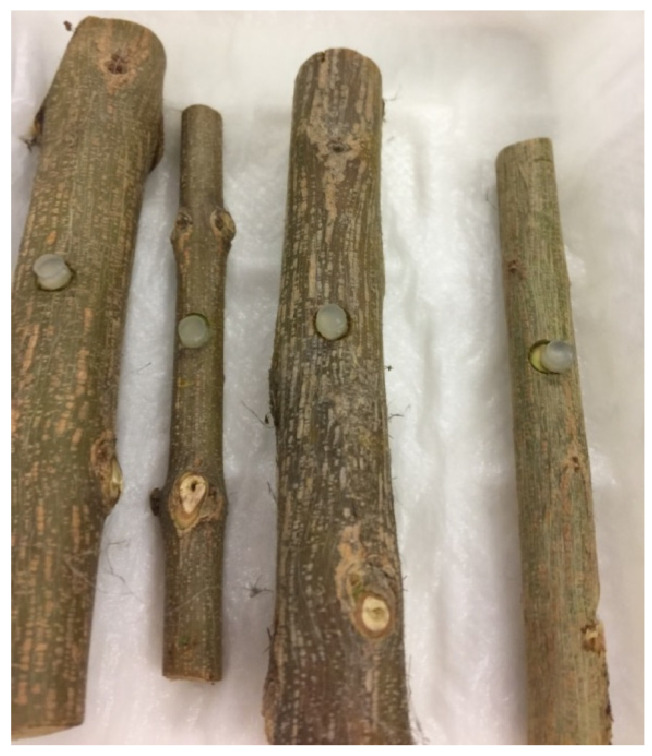
Pomegranate twigs inoculated and placed in humid chamber.

**Figure 4 jof-11-00254-f004:**
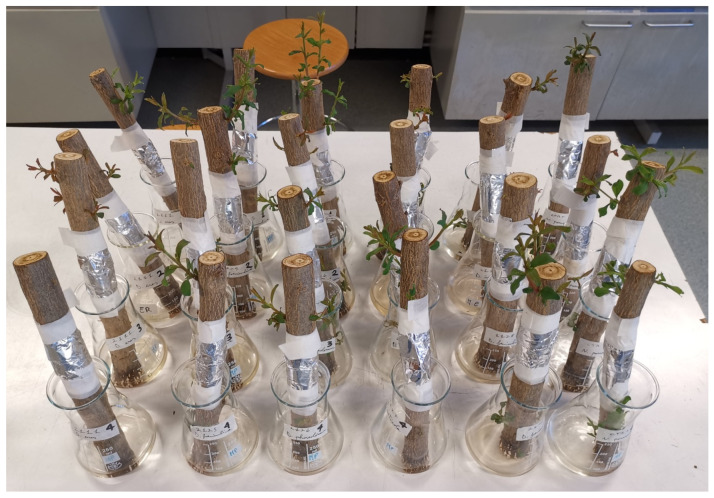
Pomegranate twigs (4 years old) inoculated and placed in water.

**Figure 5 jof-11-00254-f005:**
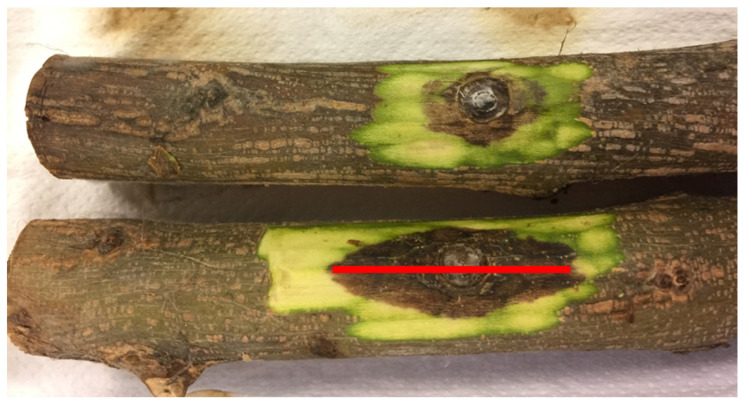
Pomegranate branches with the first layers of the bark removed to discover the browned area and measure its length in the longitudinal direction (highlighted in red).

**Figure 6 jof-11-00254-f006:**
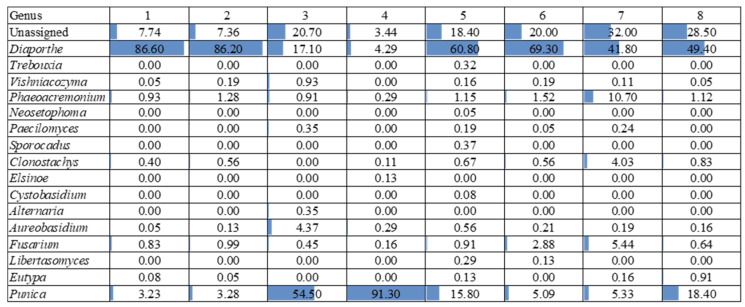
The fungal relative abundance at the genus level in the different sections of the 8 experimental units. Blue bar length within each cell graphically indicates the relative abundance. A description of each sample is provided in Table 1.

**Figure 7 jof-11-00254-f007:**
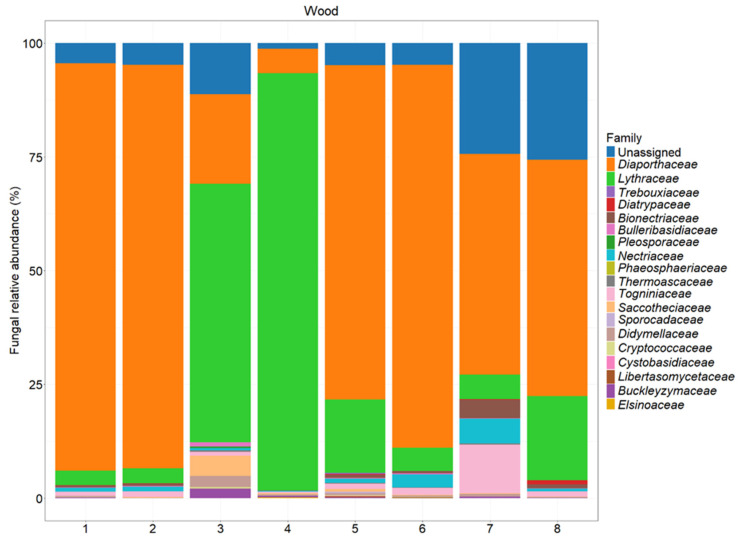
The fungal abundance at the family level in different sections of the tested plants. A description of each sample is provided in Table 1.

**Figure 8 jof-11-00254-f008:**
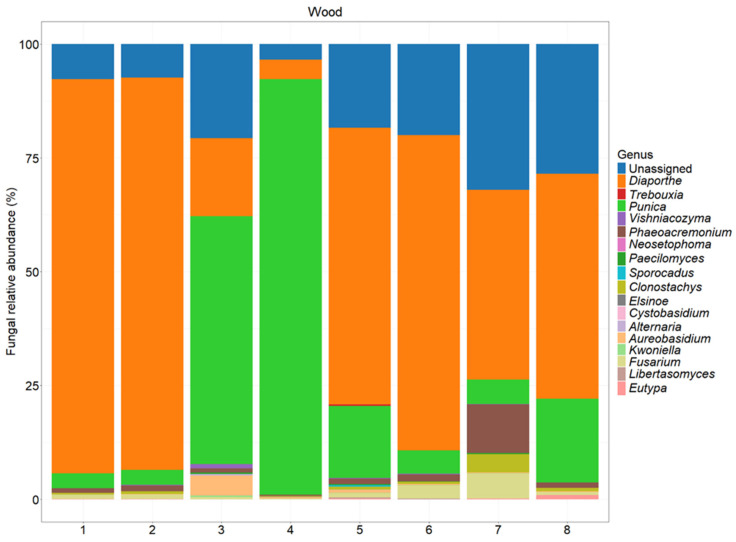
The fungal abundance at the genus level in different sections of the tested plants. A description of each sample is provided in Table 1.

**Figure 9 jof-11-00254-f009:**
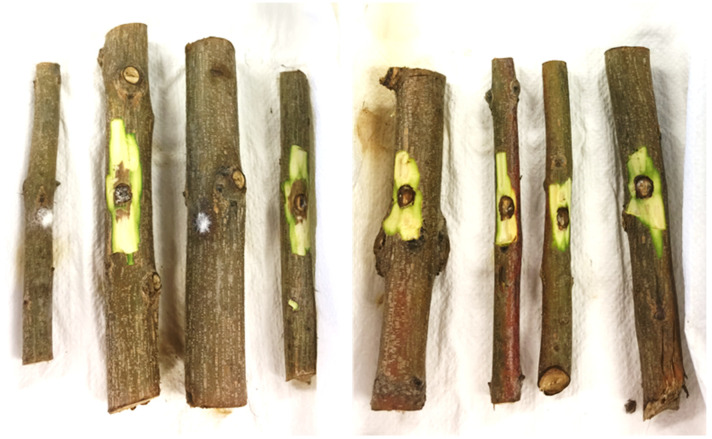
On the left (4 branches) are the results of the inoculation of *D. foeniculina* (ER 2128 tube); on the right (4 branches) are the results of the inoculation of *D. eres* (ER 2111 tube), using procedure 1.

**Figure 10 jof-11-00254-f010:**
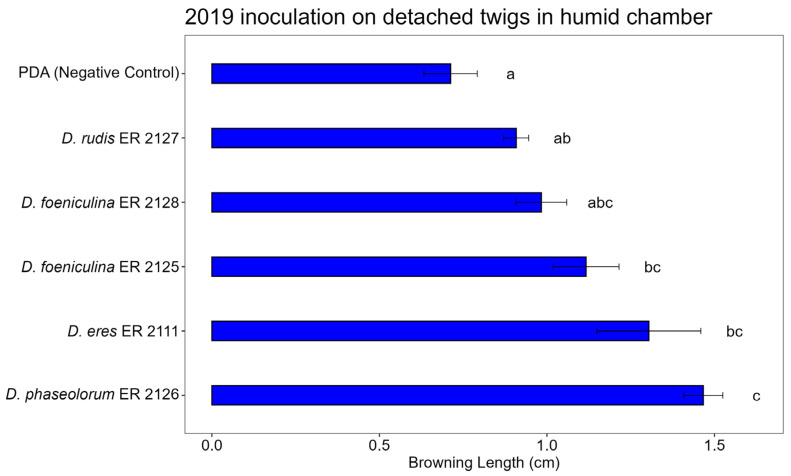
Bar plot showing average lengths (cm) of browning caused by inoculations on detached twigs placed in humid chamber is reported. Letters above bars indicate significant statistical differences between treatments (Tukey’s HSD, *p* < 0.05). Different letters indicate significant differences based on One-Way ANOVA results at *p* < 0.05. Error bars represent standard error of mean (4 replicates).

**Figure 11 jof-11-00254-f011:**
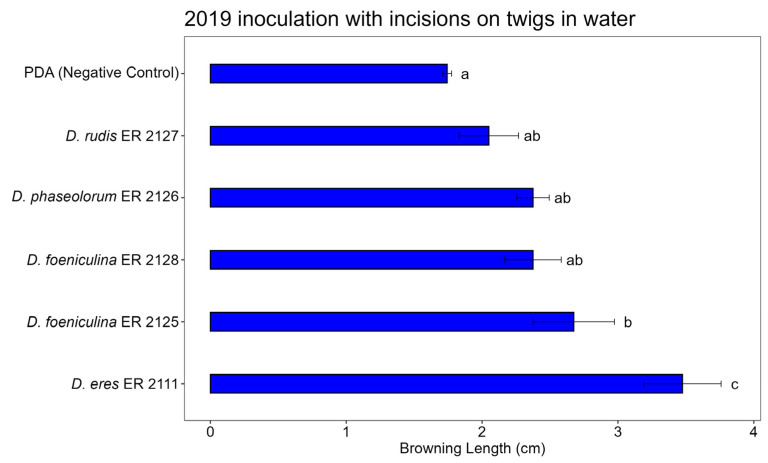
Bar plot showing average lengths (cm) of browning caused by inoculations on detached twigs in water is reported. Letters above bars indicate significant statistical differences between treatments (Tukey’s HSD, *p* < 0.05). Different letters indicate significant differences based on One-Way ANOVA results at *p* < 0.05. Error bars represent standard error of mean (4 replicates).

**Figure 12 jof-11-00254-f012:**
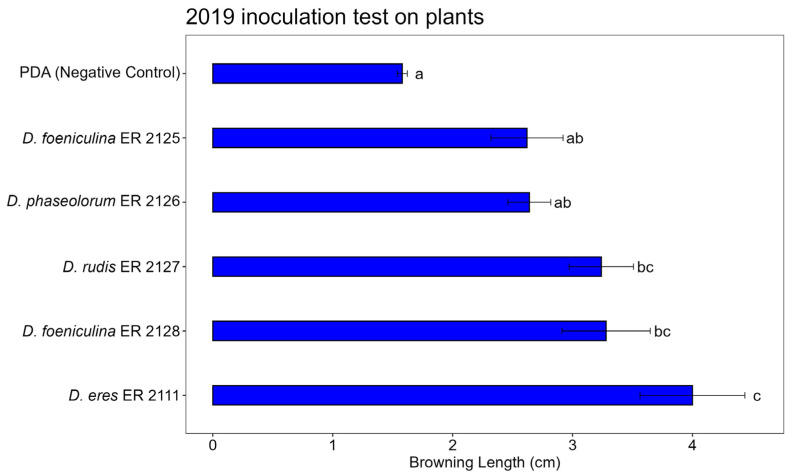
Bar plot showing average browning lengths (cm) caused by inoculations on pomegranate plants in 2020. Letters above bars indicate significant statistical differences between treatments (Tukey’s HSD, *p* < 0.05). Different letters indicate significant differences based on One-Way ANOVA results at *p* < 0.05. Error bars represent standard error of mean (4 replicates).

**Figure 13 jof-11-00254-f013:**
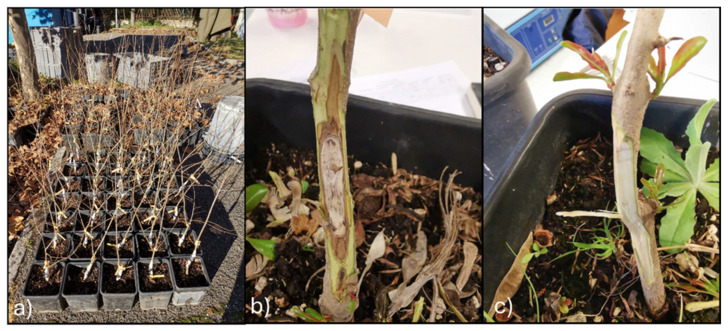
(**a**) Pomegranate plants immediately after inoculation. (**b**) A pomegranate trunk with the outer bark layers removed, revealing a browned area at the end of the trial with *D. eres*. (**c**) Negative control: a pomegranate trunk with the outer bark layers removed, inoculated with sterile PDA, showing healthy tissues.

**Table 1 jof-11-00254-t001:** A description of the experimental units is reported in Figure 2.

Experimental Units	Section	Phenotypic Features
1	1R8A + 1R8B	Brown tissues above the canker from the apical portion (Plant 1)
2	1R7A	Brown tissues from the lateral branch close to the canker (Plant 1)
3	1R2A + 1R8C	Asymptomatic tissues from the branches (Plant 1)
4	1R5A	Asymptomatic tissues (showing bark roughness) from the lateral branch (Plant 1)
5	1T1A	Brown tissues from the basal area of the trunk, corresponding to the collar (Plant 1)
6	1T1B	Brown tissues below the canker from the trunk (Plant 1)
7	1T1C + 1T1D	Brown tissues into the canker area (Plant 1)
8	2T1A + 2T1C	Brown tissues from the trunk (Plant 2)

**Table 2 jof-11-00254-t002:** List of fungal species (putative) isolated from symptomatic pomegranate wood in two sampling sites, with sampling times, representative strains, and GenBank numbers. * Samples used for metabarcoding analysis, from all 8 experimental units.

Strain CREA-DC Collection Code	Species	Isolation Incidence (% of the Total Fungal Colonies)	Pavona			Sermoneta		GenBank Accession Number
			July 2017	October 2017	March 2018 *	September 2018	February 2019	
ER2123	*N. parvum*	27.6	Absent	Present	Absent	Present	Absent	MW020287
ER2111	*D. eres*	20.7	Present	Absent	Present	Absent	Absent	MW020285
ER2125, ER2128	*D. foeniculina*	17.2	Absent	Present	Present	Absent	Absent	MW020288,MW032268
ER2126	*D. phaseolorum*	3.4	Absent	Present	Absent	Absent	Absent	MW020289
ER2127	*D. rudis*	3.4	Absent	Present	Present	Absent	Absent	MW032267
ER2145	*Cytospora acaciae*	3.4	Absent	Absent	Absent	Present	Absent	/
/	*Eutypa lata*	0.5	Absent	Absent	Present	Absent	Absent	/
ER2130	*Gliocladium* spp.	24.1	Absent	Present	Present	Present	Present	/
/	*Phoma glomerata*	5.3	Absent	Absent	Present	Absent	Present	/
/	*Neopestalotiopsis clavispora*	1	Absent	Absent	Present	Absent	Absent	/
/	*Fusarium* spp.	18.6	Absent	Absent	Present	Absent	Absent	/

**Table 3 jof-11-00254-t003:** ANOVA results conducted on length (cm) of browning caused by inoculated fungi on pomegranate-detached twigs in humid chamber in 2019.

EFFECT	SS	DF	MS	F	ProbF
Isolate	1.49215	5	0.29843	13.64701997	3.93174 × 10^−5^
Residual	0.328016667	15	0.021867778		
Total	1.94685	23	0.084645652		

**Table 4 jof-11-00254-t004:** ANOVA results conducted on length (cm) of browning caused by inoculated fungi on pomegranate-detached twigs in water in 2019.

EFFECT	SS	DF	MS	F	ProbF
Isolate	8.419153439	5	1.683831	14.04785	4.26 × 10^−6^
Residual	2.517142857	21	0.119864		
Total	10.9362963	26	0.420627		

**Table 5 jof-11-00254-t005:** ANOVA results conducted on length (cm) of browning caused by inoculated fungi on pomegranate plants in 2020.

EFFECT	SS	DF	MS	F	ProbF
Isolate	16.79066667	5	3.358133	9.631358	3.84 × 10^−5^
Residual	8.368	24	0.348667		
Total	25.15866667	29	0.86754		

## Data Availability

The original contributions presented in this study are included in the article/Appendix A. Further inquiries can be directed to the corresponding author.

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
