# Peer review of "Pomegranate Woody Mycobiota Associated with Wood Decay"

_jof, 2025, doi:10.3390/jof11040254_

Round 1
Reviewer 1 Report
This study investigates the fungal microbiota associated with symptomatic pomegranate wood using a combined approach of traditional fungal isolation and ITS2 metabarcoding analysis. This study provides a reference for the prevention and control of fungal diseases in pomegranate. However, there are still some shortcomings in this study.
1. Only one tree was collected from each orchard as a sample to surprise ITS2 metabarcoding analysis, which I think is too few.
2. I suggest the author provide the picture of Plant 1 too in page 3.
3. Some issues with textual expression
line35: "canker mortality" or "cancer mortality"?
line48: N. parvum better be full name.
This study investigates the fungal microbiota associated with symptomatic pomegranate wood using a combined approach of traditional fungal isolation and ITS2 metabarcoding analysis. This study provides a reference for the prevention and control of fungal diseases in pomegranate. However, there are still some shortcomings in this study.
1. Only one tree was collected from each orchard as a sample to surprise ITS2 metabarcoding analysis, which I think is too few.
2. I suggest the author provide the picture of Plant 1 too in page 3.
3. Some issues with textual expression
line35: "canker mortality" or "cancer mortality"?
line48: N. parvum better be full name.
Author Response
Comments 1. Only one tree was collected from each orchard as a sample to surprise ITS2 metabarcoding analysis, which I think is too few.
Response 1: We thank the reviewer for the valuable comment. Our metabarcoding approach was designed, as further check, to support the absence of Coniella granati by integrating traditional isolation from plant tissues with molecular detection. In selecting the plants for analysis, we identified a consistent number of individuals showing comparable symptoms in the orchards. Each plant was chosen as representative of the specific area of the orchard, sampling was performed on plants located along the same row, with the selected plant positioned at the centre of the orchard to minimise edge effects.
Although the metabarcoding analysis was conducted on a single plant per condition, multiple sections of each plant were analysed to verify that Coniella granati was not present in unexpected tissues or areas of the plant. Interestingly, despite the presence of symptoms, Coniella was not detected. This finding highlights the potential of combining metabarcoding with targeted sampling as a complementary tool in pathogen detection. We believe that, in the future, this approach could be effectively scaled up to broader monitoring programs.
Comment 2. I suggest the author provide the picture of Plant 1 too in page 3.
Response 2: Accepted
Comment 3. Some issues with textual expression
line35: "canker mortality" or "cancer mortality"?
Response 3: cancer mortality
Commment 4: line48: N. parvum better be full name.
Response 4: Accepted
Reviewer 2 Report
I have provided several comments in the manuscript.
Since you have cultures, I suggest performing multi-locus analysis to determine the species precisely.
Need to provide proper way of sporulating methods and add morphological data.
please check the manuscript

Author Response
Major comments
Comment 1: I have provided several comments in the manuscript.
Response 1: All suggestions have been carefully considered and corrections were made where appropriate. Thank you.
Comment 2: Since you have cultures, I suggest performing multi-locus analysis to determine the species precisely. We thank the reviewer for the helpful comment.
Response 2: We agree that species-level resolution can be challenging working only with the ITS2 region. For this reason, we will report genus-level identification and refer to putative species, considering the limitations of both metabarcoding and Sanger sequencing in distinguishing closely related species within this region.
Comment 3: Identifying the species of both Diaporthe and Neofusicoccum is based on multi-locus thus I think you can not rely only on ITS. Therefore, I suggest to stick to the generic level.
Response 3: Many thanks again for your suggestion. We have accepted it and will refer to genus-level identification throughout the revised manuscript.
Comment 4: Need to provide proper way of sporulating methods and add morphological data.
Response 4: Thank you for your comment. Added in the text the description of the morphological identification and the sporulating method.
Detail comments
Comment 5: please check the manuscript
Response 5: All suggestions have been carefully considered and corrections were made where appropriate. Thank you.
Reviewer 3 Report
Dear Authors
General the manuscript is well-structured and the information presented in a logical manner.The Introduction provides a good overview of the background of this study but Materials & Methods, Results and Discussion must be completed/corrected.
I have made 14 queries/comments throughout the manuscript. Therefore, the present draft needs revision before further process.
I have made 14 queries/comments throughout the manuscript. Therefore, the present draft needs revision before further process.

Author Response
Comment 1: I have made 14 queries/comments throughout the manuscript. Therefore, the present draft needs revision before further process.
Response 1: Thank you for all your comments and suggestions. Corrections and improvements have been added to the text.
Round 2
Reviewer 2 Report
authors have significantly improved the MS. I have noticed the term 'Basidiomycetes' in line number 448. I suggest you to change it to 'Basidiomycetous taxa'.
nothing except above comment in line 448
Author Response
Comment 1: authors have significantly improved the MS. I have noticed the term 'Basidiomycetes' in line number 448. I suggest you to change it to 'Basidiomycetous taxa'.
Response 1: Correction accepted, thank you.